# Clinical impact of pathology-proven etiology of severely stenotic aortic valves on mid-term outcomes in patients undergoing surgical aortic valve replacement

**Shiro Miura**[1]*, **Katsumi Inoue**[2☯], **Hiraku Kumamaru**[3☯], **Takehiro Yamashita**[1‡], **Michiya Hanyu**[4‡], **Shinichi Shirai**[5‡], **Kenji Ando**[5‡]

1 Department of Cardiology, Hokkaido Ohno Memorial Hospital, Sapporo, Japan, 2 Department of Laboratory Medicine, Kokura Memorial Hospital, Kitakyushu, Japan, 3 Department of Healthcare Quality Assessment, Graduate School of Medicine, The University of Tokyo, Tokyo, Japan, 4 Cardiovascular Center, Tazuke Kofukai Foundation Medical Research Institute, Kitano Hospital, Osaka, Japan, 5 Department of Cardiology, Kokura Memorial Hospital, Kitakyushu, Japan

☯ These authors contributed equally to this work.
‡ These authors also contributed equally to this work.
* s.miura@ohno-kinen.or.jp

**Data Availability Statement:** All relevant data are within the paper and its Supporting Information files.

## Abstract

### Background

The use of transcatheter or surgical aortic valve replacement (AVR) for severe aortic stenosis (AS) has considerably increased in recent years. However, the association between AS etiology and mid-term clinical outcomes after surgical AVR has not been fully investigated.

### Methods and results

We retrospectively included 201 patients (mean age, 75 years; 43%, men) who underwent surgical AVR for severe native AS (aortic valve area ≤1.0 cm$^2$ on preoperative transthoracic echocardiography examination). The following valve etiologies were postoperatively identified on pathological examination: post-inflammatory (n = 28), congenital (n = 35), and calcific/degenerative (n = 138). The median follow-up interval was 4.1 years following surgical AVR. Of the 201 patients, 27% were asymptomatic, 40% had a history of heart failure, and 11% underwent previous heart surgery. The cumulative incidence of cardiac events (all-cause death, aortic valve deterioration requiring repeated AVR, and hospitalization for heart failure) and combined adverse events, which included non-fatal stroke, unplanned coronary revascularization, pacemaker implantation, and gastrointestinal bleeding along with cardiac events, was significantly higher in the calcific/degenerative group (p = 0.02 and p = 0.02, respectively). In multivariate analysis adjusted for age, sex, renal function, heart failure, atrial fibrillation, concomitant surgical procedures, and EuroSCORE II, AS etiology was independently associated with an increased risk of combined adverse events (congenital vs. post-inflammatory: hazard ratio [HR], 4.13; p = 0.02 and calcific/degenerative vs. post-inflammatory: HR, 5.69; p = 0.002).

**Funding:** The authors received no specific funding for this work.

**Competing interests:** NO authors have competing interests.

## Conclusions

Pathology-proven AS etiology could aid in predicting the mid-term outcomes after surgical AVR, supporting the importance of accurate identification of severe AS etiology with or without postoperative pathological examination.

## Introduction

The most common form of stenotic aortic valves in Europe and the United States is calcific/degenerative, followed by those due to congenital malformations. Although rheumatic etiology is now infrequent in the West, it remains prevalent in developing countries [1]. Since the first study on the incidence of aortic valve calcification according to its etiology with increasing age in 1968 [2], these major etiologies have dominated the potential etiology of stenotic aortic valves. Transthoracic echocardiography (TTE) examination plays a central role in identifying the etiology of stenotic aortic valves by evaluating the valve appearance, number of cusps, pattern of thickening, and valve mobility [3]. However, in highly progressed aortic stenosis (AS), TTE can result in an inaccurate diagnosis, mainly because of severe aortic valve calcification or limited acoustic windows [4]. Thus, the evaluation and use of multimodality imaging, including transesophageal echocardiography, cardiac magnetic resonance imaging, and cardiac computed tomography, are highly recommended. Nevertheless, in selected cases such as congenital AS, there are still challenges in the preoperative identification of accurate AS etiology [5, 6]. When these stenotic aortic valves meet the standardized criteria for severe status in symptomatic patients irrespective of their etiology, most patients are referred to cardiovascular surgeons for surgical or transcatheter aortic valve replacement (AVR). An essential clinical implication in the identification of the accurate etiology of severe AS in patients undergoing AVR can be risk stratification by predicting its natural history, estimating surgical risk, or assessing potential comorbidities associated with its etiology. Nonetheless, there is uncertainty in the strength of the association between aortic valve etiology validated by postoperative histopathological examination and mid-term outcomes following surgical AVR.

This study aimed to (i) describe the incidence of each aortic valve etiology validated by pathological examination, highlighting the differences and similarities in clinical, echocardiographic, and operative data among AS patients with different valve etiologies requiring surgical AVR, and (ii) compare the clinical outcomes following surgical AVR with and without statistical adjustments for established risk factors.

## Materials and methods

### Study population

Among 595 consecutive patients who were initially diagnosed with severe AS with an aortic valve area (AVA) $\leq 1.0$ cm$^2$ on TTE examination [7] between August 2009 and February 2012 and followed up at our institution up to June 2015, 210 underwent surgical AVR, whereas 360 were medically managed, 18 were treated with transcatheter AVR, and follow-up data were lost for 7 (Fig 1). It is to be noted that no patient underwent other surgical aortic valve procedures, such as aortic valve repair, AVR with human homograft, or Ross procedure. Following exclusion of patients with a previous history of cardiac surgery on the aortic valve (n = 5), unavailability of pathology data (n = 2), and unidentified valve etiology (n = 2), the remaining 201 patients were included in the final analysis. Patients were divided into the following three groups according to accurate aortic valve etiology verified only by postoperative histological examinations: post-inflammatory (n = 28, 14%), congenital (n = 35, 17%), and calcific/

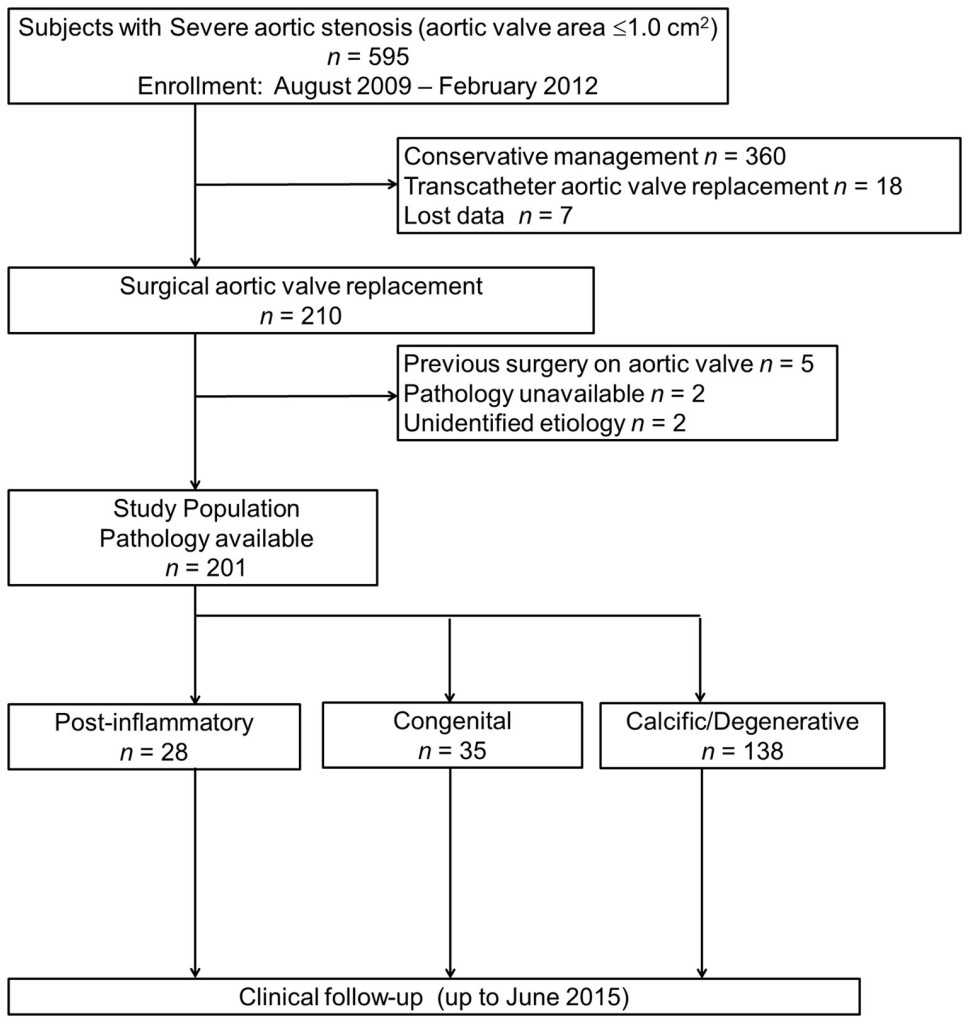

**Fig 1. Study population flow chart.**

degenerative (n = 138, 69%). Pathological features of the surgically excised aortic valves were classified based on gross and histological findings by experienced pathologists without knowledge of the patients' prognosis in 201 cases based on the main pathology criteria defined in previous reports [8–12]. Representative images from the gross examinations of the surgically excised aortic valves are presented in Fig 2. A comparison of the pathological diagnosis of the aortic valve etiology by clinical pathologists revealed a 79% (159/201 cases) concordance with the preoperative diagnosis by the attending physicians using imaging modalities (mainly TTE). Written informed consent was waived due to the retrospective nature of this study, which was reviewed and approved by the institutional review board (IRB) of Kokura Memorial Hospital. We followed the appropriate ethical protocols and privacy guidelines approved by the IRB when contacting the patients and/or their relatives.

## Clinical data

Preoperative patient characteristics included age, sex, body surface area, body mass index, symptomatic status, and laboratory data. Additionally, data on the presence of comorbidities of atrial fibrillation (AF), prior admission for heart failure (HF), previous history of cardiac

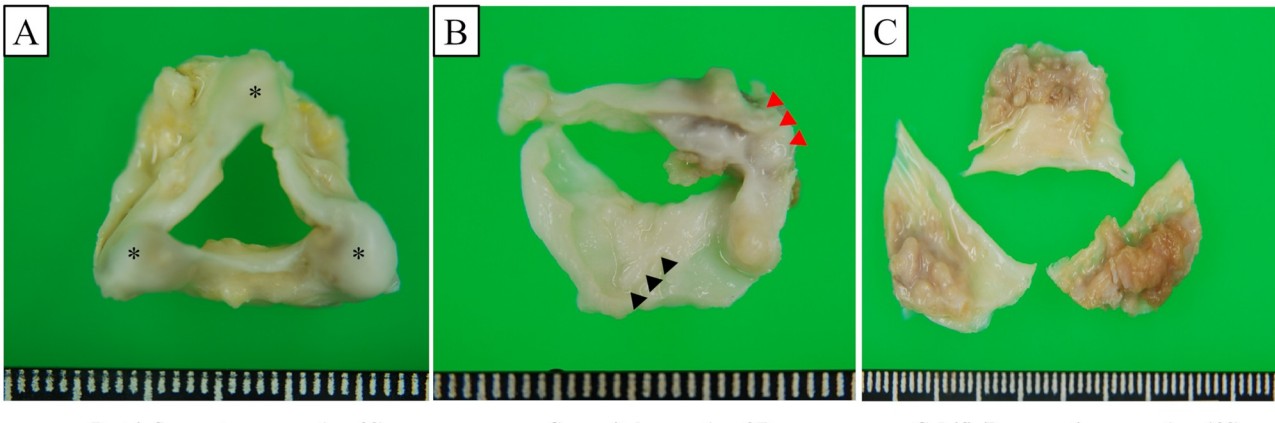

**Post-inflammatory group (n = 28)**　　**Congenital group (n = 35)**　　**Calcific/Degenerative group (n = 138)**

**Fig 2. Gross inspections of post-inflammatory (A), congenital (B), and calcific/degenerative (C) aortic stenosis in the representative cases from our study.** A: Post-inflammatory tricuspid aortic valve on the aortic side showing severe thickening of the cusps and fusion of the commissures with nodular calcification (asterisk). B: Bicuspid aortic valve on the aortic side demonstrating marked calcification of the raphe (black arrowheads) and heavy nodular calcification and severe fibrous thickening (red arrowheads). C: Calcific/degenerative tricuspid aortic valve characterized by patchy, moderate fibrotic thickening and focal heavy calcification at the base on the aortic side of the three cusps with adjacent portions of the cusps partially translucent.

surgery, and history of percutaneous coronary intervention (PCI) were collected via medical chart review. We evaluated renal function using the estimated glomerular filtration rate (GFR) calculated with the Modification of Diet in Renal Disease Study equation arranged for the Japanese population, and chronic kidney disease was diagnosed if the estimated GFR was <60 mL/min/1.73 m$^2$ or the urine albumin:creatinine ratio was >30 mg/g [13]. As an inclusive risk assessment index, both the European System for Cardiac Operative Risk Evaluation (EuroSCORE) II and the Society of Thoracic Surgeons (STS) score were calculated to compare the potential operative risk among the groups [14–16].

Comprehensive TTE was conducted by experienced sonographers using commercially available ultrasound systems (Vivid 7 Dimension and Vivid E9, GE Healthcare, Horten, Norway; iE33, Philips Medical Systems, Best, The Netherlands; Aplio SSA-700A, Toshiba Medical Systems, Tokyo, Japan) at 1 month before and after surgical AVR. Doppler echocardiographic measurements comprised the aortic mean pressure gradient using the simplified Bernoulli equation and the AVA and effective orifice area using a standard continuity equation. Stroke volume was calculated by Doppler using the velocity-time integral of the LV outflow tract and its diameter in the mid-systole of the aortic annulus in the parasternal long-axis view. For patients with AF, velocities from two or three heat cycles were averaged to reduce the influence of beat-to-beat variation. The severity of aortic, mitral, and tricuspid regurgitation was assessed following international guidelines [7].

### Definition of clinical outcome and follow-up

Clinical follow-up data after surgical AVR were obtained from medical records, patients, relatives, or attending physicians either in person or by telephone interviews until June 2015. Follow-ups were commenced on the day of index surgical AVR. Along with all-cause death as a primary endpoint, cardiac events were defined as a composite of all-cause death, aortic valve deterioration requiring repeated AVR, and hospitalization for HF. Combined adverse events included, along with cardiac events, non-fatal stroke, new-onset ischemic heart disease requiring unplanned PCI and/or coronary artery bypass grafting (CABG), symptomatic

bradycardia/ventricular tachycardia treated with pacemaker/implantable cardioverter defibrillator, and hospital admission for non-fatal gastrointestinal bleeding. Cardiovascular death was defined as deaths from worsened HF, acute coronary syndrome and stroke, postoperative in-hospital death, or sudden death. Perioperative death was defined as mortality within 30 days following surgical AVR. These definitions of the clinical outcomes were established prior to conducting this clinical study, based on our clinical interests and information from previous studies [17, 18].

## Statistical analysis

Descriptive statistics are expressed as mean ± SD or median (interquartile range), whereas categorical variables are presented as number and percentage. Differences between patients in different groups were compared using one-way analysis of variance, followed by post-hoc Bonferroni correction for continuous variables and chi-square test or Fisher's exact test for categorical variables. Kaplan-Meier curves and log-rank test for the time-to-event data were used to assess the association between the three types of aortic valve etiology and all-cause death, cardiac events, and combined adverse events; inter-group differences were tested using the log-rank test. Furthermore, the associations between aortic valve etiology and clinical outcomes were estimated using Cox proportional hazards regression models for cumulative clinical events, with adjustment for a comprehensive cardiac operative risk value (EuroSCORE II), clinically relevant variables, and risk factors consistent with previous research [19]. The proportional hazards assumption was checked using statistical tests and graphical diagnostics based on the scaled Schoenfeld residuals. A p-value $<0.05$ was considered statistically significant. All statistical analyses were performed using R statistical software version 3.3.2 (R Foundation for Statistical Computing, Vienna, Austria).

## Results

### Preoperative characteristics of the study population

A comparison of preoperative clinical characteristics and medication use at discharge is presented in Table 1. Among 201 patients (mean age, 75±9 years; 43%, men), 27% were asymptomatic, 40% had a history of HF, and 11% underwent prior open heart surgery. Patients in the calcific/degenerative group were older ($p<0.001$); had a higher prevalence of hypertension ($p<0.001$), dyslipidemia ($p = 0.03$), and coronary artery disease ($p<0.001$); had lower estimated GFR ($p = 0.002$); and were more frequently treated with antiplatelet therapy postoperatively ($p = 0.001$) than those in the other two groups. Patients in the post-inflammatory group had a higher prevalence of AF ($p<0.001$), had higher EuroSCORE II ($p = 0.003$), and were more frequently admitted for HF ($p = 0.01$). Patients in the congenital group had a higher proportion of men ($p = 0.001$) and higher body surface area on average ($p = 0.004$).

### Echocardiographic evaluations and procedural details

Periprocedural echocardiographic and surgical characteristics are summarized in Table 2. Overall, echocardiographic parameters with respect to left ventricular size and systolic function were similar in the three groups; the AVA improved from 0.66 cm$^2$ to 1.40 cm$^2$ with surgical AVR, with a drastic reduction in the mean aortic pressure gradient from 46 to 14 mmHg. Of note, patients in the post-inflammatory group were remarkable for the highest proportion of indexed stroke volume $\leq$35 mL/m$^2$ ($p = 0.03$), with moderate or severe regurgitation on the aortic ($p<0.001$), mitral ($p = 0.02$), and tricuspid ($p<0.001$) valves.

**Table 1. Preoperative characteristics according to aortic valve etiologies.**

| | Overall (n = 201) | Post-inflammatory (n = 28) | Congenital (n = 35) | Calcific/degenerative (n = 138) | p-value* |
|---|---|---|---|---|---|
| Clinical demographics | | | | | |
| Age (years) | 75 (9) | 73 (7) | 68 (13) | 76 (7) | <0.001 |
| Age >80 years | 63 (31) | 6 (21) | 7 (20) | 50 (36) | 0.09 |
| Male sex | 87 (43) | 10 (36) | 25 (71) | 52 (38) | 0.001 |
| BMI, kg/m$^2$ | 22.7 (3.3) | 21.7 (2.6) | 22.2 (2.6) | 23.1 (3.5) | 0.08 |
| BSA, m$^2$ | 1.51 (0.16) | 1.49 (0.16) | 1.59 (0.17) | 1.49 (0.16) | 0.004 |
| Asymptomatic | 54 (27) | 6 (21) | 15 (43) | 33 (24) | 0.06 |
| Hypertension | 138 (69) | 11 (39) | 19 (54) | 108 (78) | <0.001 |
| Diabetes mellitus | 57 (28) | 6 (21) | 5 (14) | 46 (33) | 0.06 |
| Dyslipidemia | 91 (45) | 10 (36) | 10 (29) | 71 (51) | 0.03 |
| Hyperuricemia | 21 (10) | 4 (14) | 3 (8.6) | 14 (10) | 0.75 |
| Prior heart failure | 81 (40) | 17 (61) | 8 (23) | 56 (41) | 0.01 |
| Coronary artery disease | 72 (36) | 5 (18) | 5 (14) | 62 (45) | <0.001 |
| Old myocardial infarction | 16 (8) | 2 (7.1) | 2 (5.7) | 12 (8.7) | 0.83 |
| Previous heart surgery | 22 (11) | 7 (25) | 2 (5.7) | 13 (9.4) | 0.03 |
| Chronic kidney disease | 170 (85) | 24 (86) | 28 (80) | 118 (86) | 0.71 |
| Hemodialysis | 20 (10) | 0 (0) | 0 (0) | 20 (15) | 0.01 |
| Chronic lung disease | 20 (10) | 3 (11) | 4 (11) | 13 (9) | 0.93 |
| Previous stroke | 22 (11) | 1 (3.6) | 3 (8.6) | 18 (13) | 0.30 |
| Atrial fibrillation | 63 (31) | 21 (75) | 5 (14) | 37 (27) | <0.001 |
| Peripheral vascular disease | 17 (8.5) | 1 (3.6) | 2 (5.7) | 14 (10) | 0.43 |
| Aortic aneurysm or dissection | 13 (6.5) | 2 (7.1) | 4 (11) | 7 (5.1) | 0.39 |
| Malignancy | 17 (8.5) | 3 (11) | 3 (8.6) | 11 (8) | 0.89 |
| Peptic ulcer disease | 11 (5.5) | 0 (0) | 3 (8.6) | 8 (5.8) | 0.32 |
| Current smoker | 13 (6.5) | 0 (0) | 8 (23) | 5 (3.6) | <0.001 |
| Medication use at discharge | | | | | |
| Antiplatelets | 94 (47) | 8 (29) | 9 (26) | 77 (56) | 0.001 |
| Beta blockers | 52 (26) | 3 (11) | 9 (26) | 40 (29) | 0.13 |
| ACEIs | 25 (12) | 3 (11) | 3 (8.6) | 19 (14) | 0.68 |
| ARBs | 82 (41) | 9 (32) | 11 (31) | 62 (45) | 0.21 |
| Statins | 80 (40) | 8 (29) | 8 (23) | 64 (46) | 0.02 |
| Calcium channel blockers | 79 (39) | 6 (21) | 14 (40) | 59 (43) | 0.11 |
| Diuretics | 71 (35) | 17 (61) | 7 (20) | 47 (34) | 0.003 |
| Anticoagulants | 46 (23) | 20 (71) | 4 (11) | 22 (16) | <0.001 |
| Laboratory data | | | | | |
| Hemoglobin (g/dL) | 12.2 (1.9) | 12.3 (2.1) | 13.6 (1.5) | 11.8 (1.7) | <0.001 |
| LDL cholesterol (mg/dL) | 110 (31) | 115 (29) | 109 (38) | 109 (30) | 0.66 |
| Uric acid (mg/dL) | 5.6 (1.8) | 5.9 (1.8) | 6.1 (2.5) | 5.4 (1.6) | 0.10 |
| Estimated GFR (mL/min/1.73 m$^2$) | 39.8 (20) | 41.5 (13) | 50.3 (17) | 36.8 (21) | 0.002 |
| BNP plasma level (pg/mL) | 148 [58, 348] | 311 [143, 469] | 81 [51, 312] | 144 [57, 305] | 0.006 |
| EuroSCORE II (%) | 2.3 [1.5, 3.8] | 2.9 [2.0, 3.3] | 1.9 [1.0, 2.4] | 2.3 [1.5, 4.1] | 0.003 |
| STS score (PROM) (%) | 3.1 [2.0, 4.2] | 3.8 [3.2, 4.7] | 1.5 [0.9, 2.3] | 3.3 [2.2, 4.3] | <0.001 |

Data are presented as mean (SD), or median [interquartile range], or n (%). BMI indicates body mass index; BSA, body surface area; ACEIs, angiotensin-converting enzyme inhibitors; ARBs, angiotensin II receptor blockers; LDL, low-density lipoprotein; GFR, glomerular filtration rate; BNP, B-type natriuretic peptide; EuroSCORE, European System for Cardiac Operative Risk Evaluation; STS, Society of Thoracic Surgeons; PROM, predicted risk of mortality.

*p-value refers to comparisons among the three groups at enrollment using analysis of variance.

**Table 2. Preoperative and postoperative echocardiographic assessments and procedural surgical AVR characteristics.**

| | Overall (n = 201) | Post-inflammatory (n = 28) | Congenital (n = 35) | Calcific/ degenerative (n = 138) | p-value[*] |
|---|---|---|---|---|---|
| Echocardiographic parameters | | | | | |
| LVEF (%) | 63 (11) | 61 (12) | 62 (13) | 63 (10) | 0.57 |
| LVEF ≤40% | 12 (6) | 3 (11) | 2 (5.7) | 7 (5.1) | 0.52 |
| LV end-diastolic dimension (mm) | 46 (6) | 46 (6) | 46 (7) | 46 (6) | 0.91 |
| LV end-systolic dimension (mm) | 30 (7) | 30 (6) | 31 (9) | 30 (6) | 0.56 |
| Aortic root diameter (mm) | 31 (4) | 31 (3) | 34 (5) | 31 (3) | <0.001 |
| Left atrial diameter (mm) | 42 (9) | 49 (13) | 38 (9) | 42 (7) | <0.001 |
| Pulmonary artery systolic pressure (mmHg) | 35 (13) | 43 (21) | 32 (11) | 33 (12) | 0.001 |
| SV (mL) | 63 (15) | 58 (12) | 66 (18) | 64 (14) | 0.05 |
| Indexed SV (mL/m$^2$) | 43 (10) | 39 (8) | 42 (11) | 43 (9) | 0.079 |
| Indexed SV ≤35 mL/m$^2$ | 45 (23) | 11 (41) | 10 (29) | 24 (18) | 0.03 |
| Preoperative AVA (cm$^2$) | 0.66 (0.15) | 0.71 (0.14) | 0.63 (0.18) | 0.66 (0.14) | 0.1 |
| Preoperative indexed AVA (cm$^2$/m$^2$) | 0.46 (0.11) | 0.47 (0.12) | 0.41 (0.11) | 0.47 (0.1) | 0.02 |
| Preoperative Vmax (m/s) | 4.4 (0.8) | 4.1 (1) | 4.7 (0.7) | 4.3 (0.8) | 0.01 |
| Preoperative MPG (mmHg) | 46 (19) | 42 (22) | 54 (17) | 45 (18) | 0.02 |
| Postoperative EOA (cm$^2$) | 1.40 (0.37) | 1.48 (0.20) | 1.51 (0.43) | 1.35 (0.36) | 0.28 |
| Postoperative Vmax (m/s) | 2.5 (0.5) | 2.3 (0.5) | 2.5 (0.6) | 2.6 (0.5) | 0.02 |
| Postoperative MPG (mmHg) | 14 (6) | 11 (5) | 14 (7) | 15 (6) | 0.04 |
| Moderate or severe AR | 17 (8.5) | 8 (29) | 2 (5.7) | 7 (5.1) | <0.001 |
| Moderate or severe MR | 20 (10) | 7 (25) | 2 (5.7) | 11 (8.0) | 0.02 |
| Moderate or severe TR | 16 (8.0) | 8 (29) | 3 (8.6) | 5 (3.6) | <0.001 |
| Surgical data | | | | | |
| Bioprosthetic valve | 160 (80) | 19 (68) | 22 (63) | 119 (86) | 0.002 |
| Mechanical valve | 41 (20) | 9 (32) | 13 (37) | 19 (14) | |
| 16 mm | 1 (0.5) | 0 (0) | 0 (0) | 1 (0.7) | 0.01 |
| 17 mm | 7 (3.5) | 1 (3.6) | 1 (2.9) | 5 (3.6) | |
| 18 mm | 1 (0.5) | 0 (0) | 0 (0) | 1 (0.7) | |
| 19 mm | 87 (43) | 12 (43) | 6 (17) | 69 (50) | |
| 20 mm | 2 (1.0) | 1 (3.6) | 1 (2.9) | 0 (0) | |
| 21 mm | 66 (33) | 9 (32) | 11 (31) | 46 (33) | |
| 22 mm | 1 (0.5) | 0 (0) | 1 (2.9) | 0 (0) | |
| 23 mm | 26 (13) | 3 (11) | 10 (29) | 13 (9.4) | |
| 24 mm | 2 (1.0) | 0 (0) | 1 (2.9) | 1 (0.7) | |
| 25 mm | 8 (4.0) | 2 (7) | 4 (11) | 2 (1.4) | |
| Concomitant procedures | 98 (49) | 23 (82) | 13 (37) | 62 (45) | 0.001 |
| CABG | 40 (20) | 1 (3.6) | 3 (8.6) | 36 (26) | 0.004 |
| Mitral valve replacement or repair | 49 (24) | 23 (82) | 2 (5.7) | 24 (17) | <0.001 |
| Tricuspid valve replacement or repair | 17 (8.5) | 10 (36) | 2 (5.7) | 5 (3.6) | <0.001 |
| Ascending aorta replacement | 13 (6.5) | 1 (3.6) | 8 (23) | 4 (2.9) | <0.001 |
| Maze operation | 4 (2) | 4 (14) | 0 (0) | 0 (0) | <0.001 |

Data are presented as mean (SD) or n (%). AVR indicates aortic valve replacement; LVEF, left ventricular ejection fraction; LV, left ventricular; SV, stroke volume; AVA, aortic valve area; Vmax, peak aortic jet velocity; MPG, mean aortic pressure gradient; EOA, effective orifice area; AR, aortic regurgitation; MR, mitral regurgitation; TR, tricuspid regurgitation; CABG, coronary artery bypass grafting.

[*]p-value refers to comparisons among the three groups using analysis of variance.

With respect to procedural details, 80% (n = 160) of patients underwent replacement with a bioprosthetic valve and 49% underwent concomitant surgical procedures. Among them, CABG was most frequently performed in the calcific/degenerative group (p = 0.004), and procedures on the mitral and tricuspid valves were the most common in the post-inflammatory group (p<0.001 each). Patients in the congenital group most frequently underwent ascending aorta replacement at 23% (p<0.001).

**Impact of the aortic valve etiology on mid-term outcome.** A comparison of detailed clinical events among the three groups is presented in Table 3. During a median follow-up period of 4.1 years (interquartile range: 2.7–5.1 years) with a 97% follow-up rate at 3 years, 30 patients (15%) died, with cardiovascular death reported in 13 (7%) of the total patients. Only one dialysis-dependent patient underwent a redo AVR performed at 3.1 years after the first AVR due to aortic valve deterioration. With respect to individual clinical events, there were no statistical differences in event rates among the three groups. The Kaplan-Meier curves for all-cause death, cardiac events, and combined adverse events are shown in Fig 3. The 3-year survival rates were 96%, 94%, and 83% for the post-inflammatory, congenital, and calcific/degenerative groups, respectively. Similarly, the 3-year survival rates free from cardiac events were 92%, 88%, and 75%, respectively, whereas the 3-year survival rates free from combined adverse events were 84%, 70%, and 64%, respectively. The cumulative incidence of both cardiac events and combined adverse events was significantly higher in the calcific/degenerative group (p = 0.02 and p = 0.02, respectively).

The Cox proportional hazards analyses of time to the three types of outcome after surgical AVR are summarized in Table 4. With respect to all-cause death, there was no association between aortic valve etiology and mortality after adjustments. However, compared to the post-inflammatory group, the calcific/degenerative group was independently associated with an increased risk of cardiac events (hazard ratio [HR], 4.45; 95% confidence interval [CI], 1.07–18.4; p = 0.04) and combined adverse events (HR, 3.59; 95% CI, 1.30–9.88; p = 0.01). After adjustments for several confounders (age, sex, previous HF, and Euro-SCORE II), the calcific/degenerative group remained independently associated with the risk of cardiac events (HR, 5.84; 95% CI, 1.39–24.4; p = 0.01). Risks for combined adverse events were significantly higher in the calcific/degenerative group (HR, 5.69; 95% CI, 1.87–17.2; p = 0.002) and congenital group (HR, 4.13; 95% CI, 1.20–14.2; p = 0.02) after adjustments for age, sex, previous HF, AF, concomitant surgical procedures, estimated GFR, and Euro-SCORE II.

The New York Heart Association functional classification of patients with severe AS at the first diagnosis, within 30 days prior to surgical AVR, and at the end of the follow-up period is summarized in Fig 4 with a comparison among the three etiologies. Eighty-six patients (43%) were asymptomatic at the first diagnosis with severe AS, which decreased to 54 (27%) preoperatively. Patients in the post-inflammatory group were more symptomatic than those in the congenital group during the preoperative period (p = 0.02). At the end of the follow-up period, 44 patients (72%) were free from cardiac symptoms, and the proportion of asymptomatic patients in the congenital group (94%) was much greater than that in the other two groups (p = 0.04).

## Discussion

Patients with severe AS due to calcific/degenerative etiology had significantly worse postoperative clinical outcomes (defined as cardiac and combined adverse events) than those with severe AS due to post-inflammatory processes who were more symptomatic preoperatively, with the most frequent admission for HF along with the highest operative risk defined by EuroSCORE

**Table 3. Detailed clinical outcomes following surgical AVR according to aortic valve etiology.**

| | Overall (n = 201) | Post-inflammatory (n = 28) | Congenital (n = 35) | Calcific/degenerative (n = 138) | p-value |
|---|---|---|---|---|---|
| All-cause death | 30 (15) | 2 (7.1) | 2 (5.7) | 26 (19) | 0.07 |
| Cardiovascular death | 13 (6.5) | 0 (0) | 2 (5.7) | 11 (7.9) | 0.29 |
| Non-cardiovascular death | 17 (8.5) | 2 (7.1) | 0 (0) | 15 (11) | 0.12 |
| Perioperative death | 7 (3.5) | 0 (0) | 0 (0) | 7 (5.1) | 0.19 |
| Hospitalization for heart failure | 22 (11) | 1 (3.6) | 4 (11) | 17 (12) | 0.40 |
| Aortic valve deterioration | 1 (0.5) | 0 (0) | 0 (0) | 1 (0.7) | 0.80 |
| Unplanned coronary revascularization | 9 (4.5) | 0 (0) | 0 (0) | 9 (6.5) | 0.12 |
| Non-fatal stroke | 13 (6.5) | 3 (11) | 0 (0) | 10 (7.3) | 0.18 |
| Pacemaker or ICD implantation | 11 (5.5) | 0 (0) | 3 (8.6) | 8 (5.8) | 0.32 |
| Gastrointestinal bleeding | 15 (7.5) | 0 (0) | 3 (8.6) | 12 (8.7) | 0.27 |
| Total follow-up period (years) | 4.1 [2.7, 5.1] | 4.0 [3.2, 4.5] | 4.6 [3.4, 5.3] | 4.1 [2.4, 5.2] | 0.22 |

Data are presented as n (%) or median [interquartile range]. AVR indicates aortic valve replacement; ICD, implantable cardioverter defibrillator

II. Likewise, the former showed poorer prognosis than the latter, although the association was not statistically significant. Patients in the congenital group had a markedly increased risk of developing combined adverse events after surgical AVR than those in the post-inflammatory group. These results highlighted the immense importance of accurate identification of the

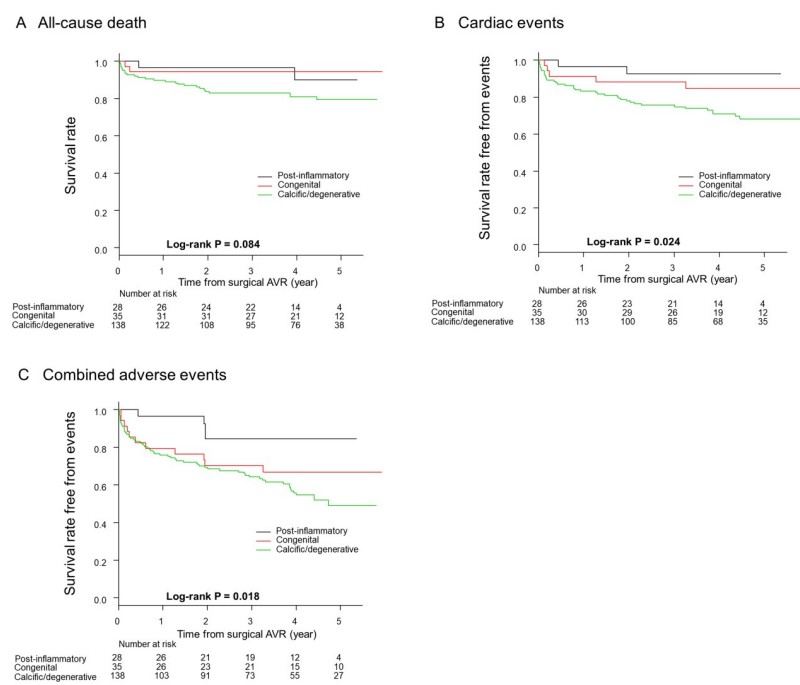

**Fig 3. Kaplan-Meier event curves for mid-term outcomes following surgical AVR among the three types of aortic valve etiologies with respect to (A) all-cause death, (B) cardiac events, and (C) combined adverse events.** Follow-ups were commenced on the day of index surgical AVR. Cardiac events were defined as all-cause death, aortic valve deterioration requiring repeated AVR, and hospitalization for HF. Combined clinical events included, along with cardiac events, non-fatal stroke, new-onset ischemic heart disease requiring unplanned coronary revascularization, symptomatic bradycardia/ventricular tachycardia treated with pacemaker/implantable cardioverter defibrillator, and admission for non-fatal gastrointestinal bleeding. AVR, aortic valve replacement.

**Table 4. Unadjusted and adjusted outcomes according to valve etiology among patients with severe AS undergoing surgical AVR.**

| Outcomes | Variables | Univariate analysis | | Multivariate analysis | |
|---|---|---|---|---|---|
| | | HR (95% CI) | p-value | HR (95% CI) | p-value |
| All-cause death | Congenital* | 0.79 (0.11–5.65) | 0.81 | 0.69 (0.09–5.40) | 0.73 |
| | Calcific/degenerative* | 2.74 (0.65–11.5) | 0.16 | 3.09 (0.73–13.1) | 0.12 |
| | Age (years) | 0.99 (0.95–1.02) | 0.63 | 0.97 (0.93–1.01) | 0.27 |
| | Male sex | 1.60 (0.78–3.29) | 0.19 | 1.73 (0.81–3.72) | 0.15 |
| | EuroSCORE II | 1.18 (1.04–1.35) | <0.01 | 1.16 (1.02–1.33) | 0.02 |
| Cardiac events | Congenital* | 2.03 (0.39–10.4) | 0.39 | 2.44 (0.44–13.5) | 0.30 |
| | Calcific/degenerative* | 4.45 (1.07–18.4) | 0.04 | 5.84 (1.39–24.4) | 0.01 |
| | Age (years) | 0.99 (0.96–1.02) | 0.79 | 0.98 (0.95–1.01) | 0.38 |
| | Male sex | 1.58 (0.89–2.81) | 0.11 | 1.87 (1.01–3.46) | 0.04 |
| | Previous heart failure | 1.76 (0.99–3.12) | 0.05 | 1.92 (1.06–3.48) | 0.03 |
| | EuroSCORE II | 1.15 (1.03–1.28) | <0.01 | 1.13 (1.01–1.26) | 0.02 |
| Combined adverse events | Congenital* | 2.39 (0.76–7.52) | 0.14 | 4.13 (1.20–14.2) | 0.02 |
| | Calcific/degenerative* | 3.59 (1.30–9.88) | 0.01 | 5.69 (1.87–17.2) | <0.01 |
| | Age (years) | 0.99 (0.97–1.02) | 0.78 | 0.98 (0.96–1.01) | 0.39 |
| | Male sex | 1.20 (0.76–1.88) | 0.41 | 1.30 (0.79–2.16) | 0.29 |
| | Previous heart failure | 1.27 (0.81–2.00) | 0.29 | 1.19 (0.74–1.92) | 0.46 |
| | Atrial fibrillation | 1.07 (0.61–1.85) | 0.80 | 1.71 (0.90–3.23) | 0.09 |
| | Concomitant surgical procedures | 1.38 (0.88–2.17) | 0.15 | 1.46 (0.90–2.37) | 0.12 |
| | Estimated GFR (mL/min/1.73 m$^2$) | 0.97 (0.96–0.98) | <0.001 | 0.98 (0.97–0.99) | 0.01 |
| | EuroSCORE II | 1.08 (0.98–1.18) | 0.09 | 1.00 (0.90–1.11) | 0.91 |

AS indicates aortic stenosis; AVR, aortic valve replacement; HR, hazard ratio; CI, confidence interval; EuroSCORE, European System for Cardiac Operative Risk Evaluation; GFR, glomerular filtration rate. Cardiac events were defined as all-cause death, aortic valve deterioration requiring repeated AVR, and hospitalization for heart failure. Combined clinical events included, along with cardiac events, non-fatal stroke, new-onset ischemic heart disease requiring unplanned coronary revascularization, symptomatic bradycardia/ventricular tachycardia treated with pacemaker/implantable cardioverter defibrillator, and admission for non-fatal gastrointestinal bleeding.

*The valve etiology "Post-inflammatory" was used as a common reference.

etiology of diseased aortic valves as well as postoperative management for comorbidities and late complications.

There are several reports on the pathology of surgically excised stenotic aortic valve; however, these involved a wide range of AS severity from mild to severe and were conducted between 1960 and 2000 [20]. Temporal changes in the etiology of stenotic aortic valves were clearly observed, with calcific/degenerative AS being the more dominant etiology in developed countries [21]. A few reports on the incidence according to aortic valve etiology have been published. In a large-scale pathology study [22] with 250 excised native stenotic aortic valves, calcific/degenerative tricuspid aortic, congenital, and rheumatic valves were reported in 66.4%, 18.4%, and 15.2% of cases, respectively, which were comparable to our findings at 68.6%, 17.4%, and 13.9%, respectively.

Echocardiography has become an established tool for the diagnosis and evaluation of aortic valve disease, and with this primary noninvasive method, aortic valve etiology has been identifiable in most cases [23]. There are, however, a limited number of cases or studies on AS that were unidentified or misled using these noninvasive imaging modalities in daily practice [24, 25]. A recently published meta-analysis study investigated a pooled sensitivity of 87.7% and pooled specificity of 88.3% for bicuspid aortic valve on preoperative TTE examinations, highlighting that suboptimal diagnosis often occurred under certain conditions, including

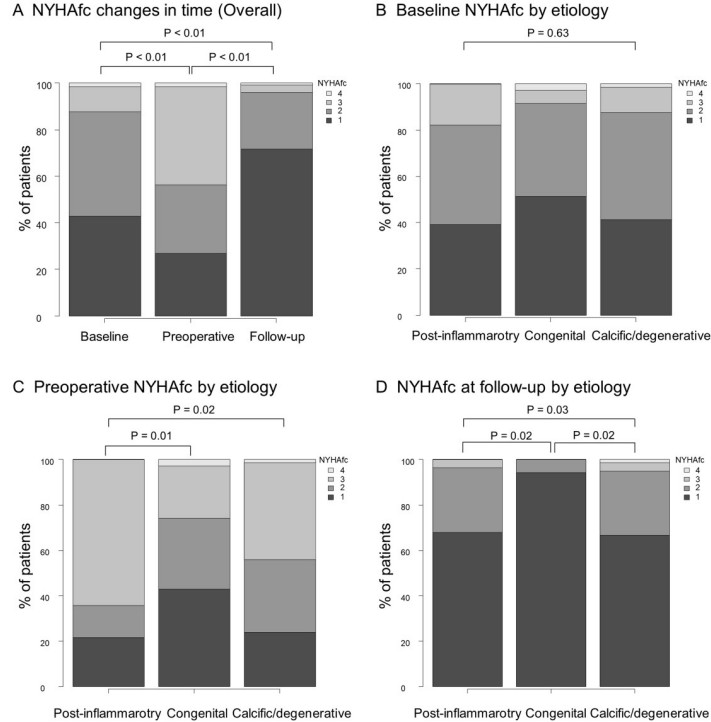

**Fig 4. NYHAfc of patients with severe aortic stenosis at the first diagnosis, within 30 days prior to surgical AVR, and at 3 years after surgical AVR (A) with a comparison of the three different aortic valve etiologies (B–D).** NYHAfc, New York Heart Association functional classification; AVR, aortic valve replacement.

non-tertiary care center, the presence of aortic aneurysm, and the presence of severe aortic valve calcification [5]. Although preoperative cardiac computerized tomography can be more accurate in differentiating between tricuspid and bicuspid aortic valves than TTE examinations, sufficient studies have not been implemented in the distinction between calcific/degenerative and post-inflammatory valves within the tricuspid aortic valves [25].

With respect to clinical outcomes, there exist limited data on whether aortic valve etiology postoperatively validated by pathological examination can affect mid-term outcomes following the surgical procedure. In a relatively large-scale study [18] with a design similar to ours, tricuspid AS was associated with a higher prevalence of cardiovascular risk factors and worse survival rates after surgical AVR compared with bicuspid AS. Our findings suggested that calcific/degenerative AS could have an association with an increased risk of cardiac and combined adverse events versus post-inflammatory AS. Based on our findings, we speculate that the valve etiology profiles might have some effects on the mid-term outcomes even after removal of the diseased valve. Such differences in late complications among the three groups may possibly stem from the extent of postoperative regression of diffuse myocardial fibrosis and myocardial cellular hypertrophy based on the aortic valve etiology, as investigated in a recent study [26]. Recently, calcific/degenerative AS has been understood to be an active disease process akin to atherosclerosis from the perspectives of compelling histopathological and clinical data [27]. In contrast, the congenital aortic valve disease often involves many vascular abnormalities [28], and the post-inflammatory disease can affect the myocardium and heart valves and the brain, joints, and skin [29]. In our present study, 23% of patients in the congenital group underwent ascending aorta replacement concomitantly, and 82% and 36% of patients in the post-inflammatory group underwent additional surgical

procedures on the mitral and tricuspid valves, respectively. Taking this into consideration, AS should be deemed as a systemic disease with different potential mechanisms by its valve etiology, although the definite pathophysiology behind AS remains incompletely investigated [30].

As a clinical implication, our data suggest that the etiology and pathobiology of the valvular disease should be examined through pathological investigation of the excised valve, which can provide essential information on its natural history, surgical risk, postoperative outcome, association with systemic disease, and potential vascular complications, irrespective of preoperative imaging assessments. Therefore, the future possibility of establishing accurate diagnosis of diseased valves using less invasive imaging modalities with pathological examination should be further explored, which can lead to other novel researches, such as valve progression by aortic valve etiology or associations with non-aortic valves. Moreover, our data on the detailed clinical outcomes highlight distinct trends in the incidences of harmful events following surgical AVR among the three etiologies. For instance, admission for HF, device implantation, and gastrointestinal bleeding were more prominent in both the congenital and calcific/degenerative groups than in the post-inflammatory group, whereas non-fatal stroke was more common in both the calcific/degenerative and post-inflammatory groups. In managing these patients, such novel information could aid in detecting and treating potential late complications unique to each valve etiology.

## Study limitations and strengths

The strength of our study lies in the larger number of enrolled patients in the cohort. We collected detailed data on patient demographics, physical and medical conditions, and surgical interventions during follow-up via medical chart review and achieved a 97% follow-up rate with a median follow-up period of 4.1 years. Nonetheless, our study also has some limitations. First, it was conducted at a single medical institute in Japan and may not represent the overall population or be applicable to other populations. Our institution is the only large cardiovascular center located in the Kitakyushu area, and almost all cardiovascular patients within the region are referred to our institution because of the lack of viable alternatives. Second, the study was mainly a descriptive one with a retrospective design. There was an imbalance in preoperative characteristics among the three etiology groups, suggesting that confounding factors might have remained even when performing a multivariate regression analysis. Moreover, owing to the limited number of outcomes, we were unable to make full adjustments even for the variables that we measured. Therefore, our results should not be considered an estimate of the causal relationship between valve etiology and outcome; rather, our results indicate that the pathologically assessed etiology of diseased valves was a strong predictor of patients' mid-term outcomes even after adjustments for patients' demographic characteristics and cardiac operative risks, which could certainly guide clinicians and patients in decision-making regarding postoperative follow-ups and support the importance of the accuracy of pathological valve diagnosis. Lastly, although the mortality rate was not statistically significant among the three groups, the present study might be underpowered to detect such survival differences, mainly because the mortality rates were relatively low. However, to the best of our knowledge, our study population is one of the largest ever reported in the surgical AVR literature focusing on the comparison of major valve etiologies.

## Conclusions

This study demonstrates that the pathology-validated etiology of stenotic aortic valves could be significantly correlated with mid-term outcomes even after surgical AVR and suggested

that, compared to post-inflammatory AS, calcific/degenerative AS was associated with increased postoperative cardiac events. Likewise, compared to post-inflammatory AS, congenital AS could be associated with higher adverse clinical event rates. Our findings reinforce the clinical significance of identifying the accurate etiology with postoperative pathological examination in risk stratification, postoperative management, and prediction of potential late complications peculiar to various etiologies.

## Supporting information

**S1 File. Dataset for all individuals.**
(XLSX)

## Author Contributions

**Conceptualization:** Shiro Miura, Katsumi Inoue, Hiraku Kumamaru.

**Data curation:** Shiro Miura.

**Formal analysis:** Shiro Miura, Hiraku Kumamaru.

**Investigation:** Shiro Miura.

**Methodology:** Shiro Miura, Hiraku Kumamaru.

**Project administration:** Takehiro Yamashita.

**Resources:** Shiro Miura, Katsumi Inoue, Michiya Hanyu.

**Supervision:** Katsumi Inoue, Hiraku Kumamaru, Takehiro Yamashita, Michiya Hanyu, Shinichi Shirai, Kenji Ando.

**Validation:** Katsumi Inoue, Takehiro Yamashita, Shinichi Shirai, Kenji Ando.

**Writing – original draft:** Shiro Miura.

**Writing – review & editing:** Hiraku Kumamaru, Takehiro Yamashita.

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
