## [Decision Letter · Decision Letter 0]

11 Sep 2019

PONE-D-19-19333

Clinical impact of pathology-proven etiology of severely stenotic aortic valves on mid-term outcomes in patients undergoing surgical aortic valve replacement

PLOS ONE

Dear Dr. Miura,

Thank you for submitting your manuscript to PLOS ONE. After careful consideration, we feel that it has merit but does not fully meet PLOS ONE’s publication criteria as it currently stands. Therefore, we invite you to submit a revised version of the manuscript that addresses the points raised during the review process.

Aside from reviewer comments, please address the small sample size of the post-inflammatory and congenital groups and how this might limit power and generalizability of the results. The three groups are very different and it is uncertain if residual confounding exists. The combined adverse outcomes are not well matched to the disease pathology and needs to be well justified.

We would appreciate receiving your revised manuscript by Oct 26 2019 11:59PM. To enhance the reproducibility of your results, we recommend that if applicable you deposit your laboratory protocols in protocols.io, where a protocol can be assigned its own identifier (DOI) such that it can be cited independently in the future. For instructions see: http://journals.plos.org/plosone/s/submission-guidelines#loc-laboratory-protocols

We look forward to receiving your revised manuscript.

Kind regards,

Wen-Chih Wu, MD, MPH

Academic Editor

PLOS ONE

Journal Requirements:

Reviewers' comments:

Reviewer's Responses to Questions

**Comments to the Author**

1. Is the manuscript technically sound, and do the data support the conclusions?

Reviewer #1: Yes

Reviewer #2: Partly

2. Has the statistical analysis been performed appropriately and rigorously? 

Reviewer #1: Yes

Reviewer #2: No

3. Have the authors made all data underlying the findings in their manuscript fully available?

Reviewer #1: Yes

Reviewer #2: Yes

4. Is the manuscript presented in an intelligible fashion and written in standard English?

Reviewer #1: Yes

Reviewer #2: Yes

5. Review Comments to the Author

Reviewer #1: This is a retrospective analysis of 201 patients undergoing aortic valve replacement (AVR) for severe stenosis, with surgical pathology dividing patients into 1) post-inflammatory, 2) congenital, or 3) calcific/degenerative

Not surpisingly, those with calcific aortic valve pathology were older, with higher number of comorbidities, notably coronary artery disease, hypertension, that portended worse outcomes on median 4.1 year follow up. Also not surprising was the finding that the congenital group was younger and a larger proportion remained asymptomatic after AVR. Not so intuitive was the finding that the congenital group had a higher incidence of combined adverse events compared with the post-inflammatory group.

This study points to the important ideologic concept that aortic valve disease is systemic disease, and accurate diagnosis implies attention to other systems that are predicted to be affected according to type.

The manuscript would be more informative if a few points were clarified:

1. Line 38 states that “There has been a considerable increase in the number of aortic valve replacements (AVR) for severe aortic stenosis (AS) in recent years. What is the evidentiary basis for this statement?

2. One of the important values of this work is the validation or not of preoperative imaging in determining which risk category to place expectations for that patient. How much should we doubt our imaging and how aggressive should we be about surgical pathology? Lines 290-3 state that discordant cases were found when disease type (based on mitral involvement) was misdiagnosed as post inflammatory aortic disease when in fact it was calcific. It would be helpful to quantify the number and character of discordant diagnoses to know the scope of accuracy in this study. The study references prior work (references 5, 23) and points out the importance of diagnostic accuracy but does not quantify the diagnostic accuracy of imaging of this study.

3. The methods section describes that these were all patients undergoing AVR, and table 2 suggests that only bioprostheses and mechanical AVRs were included. The congenital population undergoes the Ross procedure more frequently than any other type of AVR, frequently with aortic root replacement, not infrequently with Konno ventriculoplasty. The aortopathy associated with bicuspid valve predisposes to aneurysm formation and, while in this study population more congenital subjects underwent concomitant root replacement, 23% seems low. Did the study include Rosses? It would be informative to clarify

4. The clinical follow up was until June 2015. Is there a reason why the data are 5 years old?

Reviewer #2: The investigators present a single institutional, retrospective study of 201 patients who underwent surgical aortic valve replacement (AVR), from 2009-2012, comparing outcomes based on histologic pathology: post inflammatory (n=28), congenital (n=35), calcific/degenerative (n=138). A consort diagram is included that defines their study population. Significant baseline differences were seen between comparative groups. More preoperative risk factors were seen with the calcific/degenerative groups. On long-term follow-up, adjusted survival was comparable for the three groups; however, an association was seen between calcific/degenerative pathology and an increased rate of nonfatal adverse events. The authors conclude that underlying correct identification of aortic pathology is important.

Comments (including minor):

The theory (that AS should be considered a systematic disease, line 326) and the results of the study are interesting although the study itself may benefit from a more clearly defined study purpose and hypothesis. This would allow for the removal of any unnecessary data or analyses and allow a more detailed focus on the elements directly related to the study question. While the importance of conscientious pathologic examination of excised specimens - including aortic valves is intuitively understood, it's not clear how this study adds to that argument.

It appears that the study may have originally conducted to determine if certain valve pathologies were associated with worse prognosis (analogous to the prognostic value of certain cancer histologies). As the authors discuss, the initial association seen seems likely to be a surrogate given the risk-adjusted survival outcomes for the three groups was comparable. Although they did find an association between certain pathologies (degenerative namely) with adverse events, there is no clear pretext for undertaking this analysis.

The authors should consider including a mechanism to explain their rationale for including as many adverse events as they did in their analysis to avoid the appearance of retrofitting their study outcomes coincide with the results found. For example, it seems likely that the investigators would have found an association between the calcific/degenerative group and adverse cardiac events despite risk adjustment undertaken given the higher rates of CAD in this subgroup. What was the rationale for including need for PCI/CABG or hospital admission for GI bleeding as an adverse event? Did they decide on these adverse events decided prior to conducting the review?

The rationale for examining NYHA class symptoms at the time of diagnosis and just prior to surgery is not entirely clear. If the authors were interested in examining the association between functional status and valve pathology, this would be a different study question which would then benefit from a different study design in which they could more formally track the progression of symptoms and improvement in each of the groups after surgery.

I was unable to find the Cox proportional hazards regression model (Table 4) in the manuscript.

Although the manuscript is easy to follow as written, a more traditional approach is to present the written text altogether, followed by tables and a figure legend/figures at the end.

The investigators should include a statement that their study followed protocol for their institutional review board - especially if they had direct contact with patients and/or family (lines 142-144).

6. PLOS authors have the option to publish the peer review history of their article (what does this mean?). If published, this will include your full peer review and any attached files.

Reviewer #1: Yes: David Bichell

Reviewer #2: No

---

## [Author Response · Author response to Decision Letter 0]

29 Oct 2019

Response to Reviewers

PONE-D-19-19333

Clinical impact of pathology-proven etiology of severely stenotic aortic valves on mid-term outcomes in patients undergoing surgical aortic valve replacement

PLOS ONE

Aside from reviewer comments, please address the small sample size of the post-inflammatory and congenital groups and how this might limit power and generalizability of the results. The three groups are very different and it is uncertain if residual confounding exists. The combined adverse outcomes are not well matched to the disease pathology and needs to be well justified.

Response: We acknowledge that two of the groups (the post-inflammatory and congenital groups) showed relatively small sample sizes compared to the other, which might affect its generalizability of our findings to some extent. However, to the best of our knowledge, this clinical study, consisting of more than 200 subjects, is one of the largest populations in cohort studies dealing with major aortic valve etiologies accurately diagnosed with pathological examinations. Therefore, we added the following sentence in the discussion section: “In particular, the clinical outcomes among the post-inflammatory and congenital groups were not statistically significant because of the relatively small size of the two groups, leading to potentially low statistical power.” (Page 27, Line 380). Interestingly, the breakdown of the three etiologies in our study were expectedly unbalanced, with calcific and degenerative AS more common and patient profiles by aortic valve etiology were considerably different in both patient backgrounds and echocardiographic parameters. We believe that those findings were one of the most important results in our study, highlighting a comprehensive message that aortic valve etiology was significantly related to its patient characteristics and more dominated by calcific and degenerative AS in the contemporary era. In addition, as the editor pointed out, there was imbalance in the preoperative characteristics across the three etiology groups, suggesting that even with multivariable regression analysis, confounding may remain. Also, due to the limited number of outcomes, we were unable to make full adjustments even for the variables that we measured. Therefore, our results should not be considered as an estimate of a causal relationship between valve etiology and outcome, but as showing that the pathologically assessed etiology of the diseased valve was a strong predictor of mid- term outcome of the patient even after basic adjustments for patient demographics and perioperative risks (represented by Euroscore II). This could certainly guide clinicians and patients in making decisions about post-operative follow-ups and support the importance of the accuracy of pathological valve diagnosis. To clarify this, we have made adjustments to the discussion section as follows: “Thirdly, there was an imbalance in the preoperative characteristics across the three etiology groups, suggesting that confounding factors may remain even on performing multivariable regression analysis. Moreover, due to the limited number of outcomes, we were unable to make full adjustments even for the variables that we measured. Therefore, our results should not be considered an estimate of a causal relationship between valve etiology and outcome; rather, our results show that the pathologically assessed etiology of the diseased valve was a strong predictor of the mid-term outcomes of the patient even after adjusting for patient demographic characteristics and perioperative risks (represented by EuroSCORE II). This could certainly guide clinicians and patients in decision-making regarding the postoperative follow-ups and support the importance of the accuracy of pathological valve diagnosis.” (Page 26, Line 369) 

Regarding combined adverse outcomes, we investigated the detailed clinical outcomes to make a comparison among the three groups in table 3. Intriguingly, there was a trend that patients in the Calcific/degenerative group developed adverse cardiovascular events and gastrointestinal bleeding more frequently than others. Overall, we assumed that those differences in event rates could lead to a statistical difference of both cardiac events and combined adverse events found in Table 4 and Figure 3. From an appropriate statistical standpoint, we decided prior to conducting the clinical study to focus on these three types of clinically harmful events (all-cause death, cardiac events [all-cause death, aortic valve deterioration requiring repeated AVR, and HF hospitalization] and combined adverse events [cardiac events, non-fatal stroke, new-onset ischemic heart disease requiring unplanned PCI, and/or CABG, symptomatic bradycardia/ventricular tachycardia treated with pacemaker/ implantable cardioverter defibrillator, and hospital admission for non-fatal gastrointestinal bleeding]) for which we had a clinical interest. Additionally, these three adverse events have been commonly used as endpoints for patients undergoing interventions on aortic valve in recently published papers [1, 2]. Thus, we never retrofitted our data after analyzing the original data with prespecified multivariate analysis. We found a higher incidence of heart failure admission and late GI bleeding following successful surgical AVR for various mechanisms [3], both of which sometimes resulted in more fatal consequences than expected. A number of previous studies have linked aortic valve stenosis to GI bleeding [4] and reported cessation of GI bleeding following surgical AVR [5]. Moreover, in the new era of transcatheter AVR, there is growing attention to coronary revascularization strategies involving whether complete revascularization should be done along with surgical AVR or whether it should be accomplished prior to transcatheter AVR [2]. In our study all patients had complete revascularization performed with concomitant CABG when indicated. Therefore, including unplanned coronary revascularization in adverse events was considered relevant and met our clinical interest. Thus, we put the following sentence in the materials and methods section: “These definitions of the clinical outcomes were established prior to conducting this clinical study, based on our clinical interests and information from previous studies.” (Page 10, Line 162) and in the discussion section to reinforce informative arguments we demonstrated the similar study [6] to ours by adding the following sentence: “In a relatively large-scale study with a design similar to ours, tricuspid AS was associated with a higher prevalence of cardiovascular risk factors and worse survival rates after surgical AVR compared with bicuspid AS. This recent study, however, excluded patients with inflammatory AS. The Kaplan-Meier survival curves were adjusted only for age, and assessments of clinical events were not described, except for the mortality rates.” (Page 24, Line 321) 

Reviewer #1: This is a retrospective analysis of 201 patients undergoing aortic valve replacement (AVR) for severe stenosis, with surgical pathology dividing patients into 1) post-inflammatory, 2) congenital, or 3) calcific/degenerative. Not surprisingly, those with calcific aortic valve pathology were older, with higher number of comorbidities, notably coronary artery disease, hypertension, that portended worse outcomes on median 4.1 year follow up. Also, not surprising was the finding that the congenital group was younger, and a larger proportion remained asymptomatic after AVR. Not so intuitive was the finding that the congenital group had a higher incidence of combined adverse events compared with the post-inflammatory group. This study points to the important ideologic concept that aortic valve disease is systemic disease, and accurate diagnosis implies attention to other systems that are predicted to be affected according to type. The manuscript would be more informative if a few points were clarified:

Response: We highly appreciated the comment above. 

1. Line 38 states that “There has been a considerable increase in the number of aortic valve replacements (AVR) for severe aortic stenosis (AS) in recent years. What is the evidentiary basis for this statement?

Response: According to changes in annual surgery procedure volumes for aortic valve and thoracic aortic procedures from 2006 to 2016 in the U.S. [7], the significant rise in the number of aortic valve replacements has resulted from the rapid growth of transcatheter AVR. In contrast, this recent trend in surgical AVR volume is in marked contrast to the meteoric rise in the number of transcatheter AVR cases since the United States Food and Drug Administration approval the procedure in 2011. A similar upward trend has also been found in other European countries [8, 9]. To avoid misleading the reader, we edited the sentence: “There has been a considerable increase in the number of aortic valve replacements (AVR) for severe aortic stenosis (AS) in recent year” to the following one in the abstract section: “The use of a transcatheter or surgical aortic valve replacement (AVR) for severe aortic stenosis (AS) has considerably increased in recent years.” (Page 3, Line 39) 

2. One of the important values of this work is the validation or not of preoperative imaging in determining which risk category to place expectations for that patient. How much should we doubt our imaging and how aggressive should we be about surgical pathology? Lines 290-3 state that discordant cases were found when disease type (based on mitral involvement) was misdiagnosed as post inflammatory aortic disease when in fact it was calcific. It would be helpful to quantify the number and character of discordant diagnoses to know the scope of accuracy in this study. The study references prior work (references 5, 23) and points out the importance of diagnostic accuracy but does not quantify the diagnostic accuracy of imaging of this study.

Response: The comments above are highly respected because the motivation for our research originated from insights into the diagnostic accuracy of valve etiology with conventional imaging modalities. As pointed out, discordant cases were most commonly found in our study when disease type based on mitral involvement was misdiagnosed as post inflammatory aortic disease when, in fact, it was calcific. Here, we investigated the diagnostic accuracy of aortic valve etiology with preoperative imaging modalities (mainly transthoracic echocardiography) for pathology-verified diagnosis in this study where the overall concordance rate was 79% (159/201 cases). We were ready to discuss the diagnostic accuracy of imaging modality and its practical implication to improve the preoperative diagnosis. However, this interesting topic was a bit beyond the scope of this paper as this study aimed at mainly exploring the clinical impact of aortic valve etiology for predefined postoperative outcomes. As a result, we added the following sentence to describe the validity of our preoperative diagnosis: “A comparison of the pathological diagnosis of the aortic valve etiology by clinical pathologists revealed a 79% (159/201 cases) concordance with the preoperative diagnosis by the attending physicians using imaging modalities (mainly TTE)“ (Page 7, Line 108) and “The diagnostic accuracy reported in the aforementioned study was comparable to our results in that the preoperative diagnosis was in agreement with the pathologically determined etiology in approximately 80% of the cases.” in the discussion section (Page 22, Line 300)

3. The methods section describes that these were all patients undergoing AVR, and table 2 suggests that only bioprostheses and mechanical AVRs were included. The congenital population undergoes the Ross procedure more frequently than any other type of AVR, frequently with aortic root replacement, not infrequently with Konno ventriculoplasty. The aortopathy associated with bicuspid valve predisposes to aneurysm formation and, while in this study population more congenital subjects underwent concomitant root replacement, 23% seems low. Did the study include Rosses? It would be informative to clarify.

Response: No patients in the present study underwent a Ross procedure or Konno ventriculoplasty. We considered that this is mainly because our patient population was older, with a mean age of 75 years old, for appropriate indication of the Ross procedure which is more complex and technically demanding than standard AVR. Consequently, it requires a longer cardiopulmonary bypass time with the involvement of another valve triggering additional risk of complications. Therefore, we put the following sentences in the method section for clarification: “It is to be noted that no patient underwent other surgical aortic valve procedures, such as aortic valve repair, aortic valve replacement with human homograft, or the Ross procedure.” (Page 6, Line 98) 

4. The clinical follow up was until June 2015. Is there a reason why the data are 5 years old?

Response: In truth, I was studying overseas to obtain a master’s degree in medical statics at the University of Southampton for almost two years. Following completion of that program and graduation, I returned to Japan and investigated this novel paper using medical knowledge and statistical skills with clinical data obtained between 2009 and 2015. We believe this period of time is ideal for this particular study because transcatheter AVR procedures, which had never provided information on valve pathology, had just been officially approved by Japanese FDA for use in 2013. 

Reviewer #2: The investigators present a single institutional, retrospective study of 201 patients who underwent surgical aortic valve replacement (AVR), from 2009-2012, comparing outcomes based on histologic pathology: post inflammatory (n=28), congenital (n=35), calcific/degenerative (n=138). A consort diagram is included that defines their study population. Significant baseline differences were seen between comparative groups. More preoperative risk factors were seen with the calcific/degenerative groups. On long-term follow-up, adjusted survival was comparable for the three groups; however, an association was seen between calcific/degenerative pathology and an increased rate of nonfatal adverse events. The authors conclude that underlying correct identification of aortic pathology is important.

Comments (including minor):

The theory (that AS should be considered a systematic disease, line 326) and the results of the study are interesting although the study itself may benefit from a more clearly defined study purpose and hypothesis. This would allow for the removal of any unnecessary data or analyses and allow a more detailed focus on the elements directly related to the study question. While the importance of conscientious pathologic examination of excised specimens - including aortic valves is intuitively understood, it's not clear how this study adds to that argument. It appears that the study may have originally conducted to determine if certain valve pathologies were associated with worse prognosis (analogous to the prognostic value of certain cancer histologies). As the authors discuss, the initial association seen seems likely to be a surrogate given the risk-adjusted survival outcomes for the three groups was comparable. Although they did find an association between certain pathologies (degenerative namely) with adverse events, there is no clear pretext for undertaking this analysis.

Response: We wholly appreciate the comments from the reviewer. First, we need to clarify our study aim and its implications for clinical practice. A recent study [10] showed that the progression of AS in both tricuspid (post-inflammatory and calcific and degenerative AS) and congenital AS shares a common disease process of inflammation, calcium deposition and ossification and those three entities were thought to be a systemic disease. Although the relief of severe stenotic valves via surgical replacement is associated with improved outcomes over medical management, there is limited data on investigating the effects of aortic valve etiology on clinical outcomes following surgical AVR. Furthermore, potential differences among the three groups, such as burden of associated cardiovascular risk factors, associated aortopathies or differences in cardiac structures might influence postoperative outcomes and help manage patients [11]. Second, the study was mainly a descriptive one, and while we sought to compare predefined clinical outcomes among the three groups adjusting for as many potential confounding variables as possible, our study does not aim to imply inferiority or superiority of a certain valve etiology with respect to postoperative outcomes because few studies exist comparing the three AS etiologies. Needless to say, these estimated hazard ratios should not be interpreted as causal effects of the predictors to clinical outcomes of the patients. To wrap up, the ultimate goal of this study is to (i) describe the incidence of each valve etiology highlighting differences and similarities in clinical, echocardiographic and operative data among the three groups and (ii) compare clinical outcomes across the three etiologies following surgical AVR with and without statistical adjustments. Accordingly, we changed the sentences: “We aimed to investigate the association between the etiologies of stenotic aortic valves and clinical outcomes after surgical AVR in patients with severe AS.” in the introduction to the following:” This study aimed to (i) describe the incidence of each aortic valve etiology verified by pathological examinations, highlighting differences and similarities in clinical, echocardiographic, and operative data among AS patients with different valve etiologies requiring surgical AVR, and (ii) compare the clinical outcomes following surgical AVR with and without statistical adjustments for the established risk factors.”(Page 6, Line 85) 

The authors should consider including a mechanism to explain their rationale for including as many adverse events as they did in their analysis to avoid the appearance of retrofitting their study outcomes coincide with the results found. For example, it seems likely that the investigators would have found an association between the calcific/degenerative group and adverse cardiac events despite risk adjustment undertaken given the higher rates of CAD in this subgroup. What was the rationale for including need for PCI/CABG or hospital admission for GI bleeding as an adverse event? Did they decide on these adverse events decided prior to conducting the review?

Response: From an appropriate statistical standpoint, we decided prior to conducting the clinical study to focus on these three types of clinically harmful events (all-cause death, cardiac events [all-cause death, aortic valve deterioration requiring repeated AVR, and HF hospitalization] and combined adverse events [cardiac events, non-fatal stroke, new-onset ischemic heart disease requiring unplanned PCI, and/or CABG, symptomatic bradycardia/ventricular tachycardia treated with pacemaker/ implantable cardioverter defibrillator, and hospital admission for non-fatal gastrointestinal bleeding]) for which we had a clinical interest. Additionally, these three adverse events have been commonly used as endpoints for patients undergoing interventions on aortic valve in recently published papers [1, 2]. Thus, we never retrofitted our data after analyzing the original data with prespecified multivariate analysis. We found a higher incidence of heart failure admission and late GI bleeding following successful surgical AVR for various mechanisms [3], both of which sometimes resulted in more fatal consequences than expected. A number of previous studies have linked aortic valve stenosis to GI bleeding [4] and reported cessation of GI bleeding following surgical AVR [5]. Moreover, in the new era of transcatheter AVR, there is growing attention to coronary revascularization strategies involving whether complete revascularization should be done along with surgical AVR or whether it should be accomplished prior to transcatheter AVR [2]. In our study all patients had complete revascularization performed with concomitant CABG when indicated. Therefore, including unplanned coronary revascularization in adverse events was considered relevant and met our clinical interest. Thus, we put the following sentence in the materials and methods section: “These definitions of the clinical outcomes were established prior to conducting this clinical study, based on our clinical interests and information from previous studies.” (Page 10, Line 162)

The rationale for examining NYHA class symptoms at the time of diagnosis and just prior to surgery is not entirely clear. If the authors were interested in examining the association between functional status and valve pathology, this would be a different study question which would then benefit from a different study design in which they could more formally track the progression of symptoms and improvement in each of the groups after surgery.

Response: The comments above are to the point. We need to admit that comparing NYHA class symptoms among three groups in Figure 4 is relative less important and a bit off our initial study purpose. When focusing on the association between valve etiology and functional status, we should design a different study as pointed out here. The main aim of describing functional status and its changes over time was to highlight how much functional status was improved by surgical procedures and maintained at follow-up regardless of aortic valve etiology. In addition, we were determined to report functional status because the development of cardiac symptoms is a clear indication for intervention on severe AS. Most essential is that we contemplated that assessing the chronic functional status at baseline, post-surgery and follow-up by aortic valve etiology could help understand what happened in patients with severe AS undergoing surgical AVR while providing different perspectives from the predefined clinical events. This is because clinical adverse events in the study was interpreted as an acute event and assessing the functional status was evaluated as a chronic status during a regular visit.

I was unable to find the Cox proportional hazards regression model (Table 4) in the manuscript.

Response: We added Table 4 to clearly present the associations between aortic valve etiology and clinical outcomes. (Page 20)

Although the manuscript is easy to follow as written, a more traditional approach is to present the written text altogether, followed by tables and a figure legend/figures at the end.

Response: We have followed submission guidelines of PLOS ONE Journal which shows the following statements regarding tables and figures: 

Do not include figures in the main manuscript file. Each figure must be prepared and submitted as an individual file.

Figure captions must be inserted in the text of the manuscript, immediately following the paragraph in which the figure is first cited (read order). Do not include captions as part of the figure files themselves or submit them in a separate document.

Place each table in your manuscript file directly after the paragraph in which it is first cited (read order). Do not submit your tables in separate files.

The investigators should include a statement that their study followed protocol for their institutional review board - especially if they had direct contact with patients and/or family (lines 142-144).

Response: The comment is valued as one of most essential parts in a clinical study dealing with humans. Consequently, we added the following sentences to show we followed protocol for our institutional review board when contacting patients and/or their relatives: “We followed the appropriate ethical protocols and privacy guidelines approved by the IRB when contacting the patients and/or their relatives.” (Page 7, Line 113)

 

References

1. Auensen A, Hussain AI, Bendz B, Aaberge L, Falk RS, Walle-Hansen MM, et al. Morbidity outcomes after surgical aortic valve replacement. Open Heart. 2017;4(1):e000588. doi: 10.1136/openhrt-2017-000588. PubMed PMID: 28674629; PubMed Central PMCID: PMCPMC5471875.

2. Sondergaard L, Popma JJ, Reardon MJ, Van Mieghem NM, Deeb GM, Kodali S, et al. Comparison of a Complete Percutaneous versus Surgical Approach to Aortic Valve Replacement and Revascularization in Patients at Intermediate Surgical Risk: Results from the Randomized SURTAVI Trial. Circulation. 2019. doi: 10.1161/CIRCULATIONAHA.118.039564. PubMed PMID: 31476897.

3. Iyengar A, Sanaiha Y, Aguayo E, Seo YJ, Dobaria V, Toppen W, et al. Comparison of Frequency of Late Gastrointestinal Bleeding With Transcatheter Versus Surgical Aortic Valve Replacement. Am J Cardiol. 2018;122(10):1727-31. doi: 10.1016/j.amjcard.2018.07.047. PubMed PMID: 30316451.

4. Shoenfeld Y, Eldar M, Bedazovsky B, Levy MJ, Pinkhas J. Aortic stenosis associated with gastrointestinal bleeding. A survey of 612 patients. Am Heart J. 1980;100(2):179-82. doi: 10.1016/0002-8703(80)90113-1. PubMed PMID: 6967691.

5. Scheffer SM, Leatherman LL. Resolution of Heyde's syndrome of aortic stenosis and gastrointestinal bleeding after aortic valve replacement. Ann Thorac Surg. 1986;42(4):477-80. doi: 10.1016/s0003-4975(10)60563-2. PubMed PMID: 3490235.

6. Huntley GD, Thaden JJ, Alsidawi S, Michelena HI, Maleszewski JJ, Edwards WD, et al. Comparative study of bicuspid vs. tricuspid aortic valve stenosis. Eur Heart J Cardiovasc Imaging. 2018;19(1):3-8. doi: 10.1093/ehjci/jex211. PubMed PMID: 29029001.

7. D'Agostino RS, Jacobs JP, Badhwar V, Fernandez FG, Paone G, Wormuth DW, et al. The Society of Thoracic Surgeons Adult Cardiac Surgery Database: 2018 Update on Outcomes and Quality. Ann Thorac Surg. 2018;105(1):15-23. doi: 10.1016/j.athoracsur.2017.10.035. PubMed PMID: 29233331.

8. Beckmann A, Meyer R, Lewandowski J, Frie M, Markewitz A, Harringer W. German Heart Surgery Report 2017: The Annual Updated Registry of the German Society for Thoracic and Cardiovascular Surgery. Thorac Cardiovasc Surg. 2018;66(8):608-21. doi: 10.1055/s-0038-1676131. PubMed PMID: 30508866.

9. Bartus K, Sadowski J, Litwinowicz R, Filip G, Jasinski M, Deja M, et al. Changing trends in aortic valve procedures over the past ten years-from mechanical prosthesis via stented bioprosthesis to TAVI procedures-analysis of 50,846 aortic valve cases based on a Polish National Cardiac Surgery Database. J Thorac Dis. 2019;11(6):2340-9. doi: 10.21037/jtd.2019.06.04. PubMed PMID: 31372271; PubMed Central PMCID: PMCPMC6626813.

10. Wallby L, Janerot-Sjoberg B, Steffensen T, Broqvist M. T lymphocyte infiltration in non-rheumatic aortic stenosis: a comparative descriptive study between tricuspid and bicuspid aortic valves. Heart. 2002;88(4):348-51. doi: 10.1136/heart.88.4.348. PubMed PMID: 12231589; PubMed Central PMCID: PMCPMC1767380.

11. Masri A, Kalahasti V, Alkharabsheh S, Svensson LG, Sabik JF, Roselli EE, et al. Characteristics and long-term outcomes of contemporary patients with bicuspid aortic valves. J Thorac Cardiovasc Surg. 2016;151(6):1650-9 e1. doi: 10.1016/j.jtcvs.2015.12.019. PubMed PMID: 26825434.

---

## [Decision Letter · Decision Letter 1]

6 Dec 2019

PONE-D-19-19333R1

Clinical impact of pathology-proven etiology of severely stenotic aortic valves on mid-term outcomes in patients undergoing surgical aortic valve replacement

PLOS ONE

Dear Dr. Miura,

Thank you for submitting your manuscript to PLOS ONE. After careful consideration, we feel that it has merit but does not fully meet PLOS ONE’s publication criteria as it currently stands. Therefore, we invite you to submit a revised version of the manuscript that addresses the points raised during the review process.

Please provide point to point responses to additional comments by reviewer 2 (below).

Please ask for a professional English editor to review the manuscript for grammar and readability prior to the next submission as minor grammar errors are present.

We would appreciate receiving your revised manuscript by Jan 20 2020 11:59PM. To enhance the reproducibility of your results, we recommend that if applicable you deposit your laboratory protocols in protocols.io, where a protocol can be assigned its own identifier (DOI) such that it can be cited independently in the future. For instructions see: http://journals.plos.org/plosone/s/submission-guidelines#loc-laboratory-protocols

We look forward to receiving your revised manuscript.

Kind regards,

Wen-Chih Wu, MD, MPH

Academic Editor

PLOS ONE

Reviewers' comments:

2. Is the manuscript technically sound, and do the data support the conclusions?

Reviewer #2: Partly

3. Has the statistical analysis been performed appropriately and rigorously? 

Reviewer #2: Yes

4. Have the authors made all data underlying the findings in their manuscript fully available?

Reviewer #2: Yes

5. Is the manuscript presented in an intelligible fashion and written in standard English?

Reviewer #2: Yes

6. Review Comments to the Author

Reviewer #2: The authors present a revision of their retrospective cohort study examining the association of valve pathology with intermediate outcome of patients undergoing surgical aortic valve replacement in which they found an association between valve pathology (mainly calcific disease) and outcome that persisted after risk adjustment.

Comments

Their revision and written communication addresses some of the comments/questions raised by reviewers. They are careful to note their association between valve type is not causal. The clinical impact of this investigation is difficult to ascertain.

The authors might include additional discussion on their interpretation of results – i.e. their interpretation of findings. They may have an idea on why, after risk adjustment, patients with calcific/degenerative pathology have worse outcomes (fatal and nonfatal complications) relative to patients with inflammatory pathology - especially given the strong association seen with adverse outcomes

Consider including the results of univariate analysis for the candidate variables use for multivariate

analysis used in Talble 4 (i.e.,, age, sex,....euroscore, etc)

Also, In the discussion, the authors might briefly discuss recommendations on how their study findings should impact clinical practice.

Figures 3 and 4 are difficult to read. Consider reformatting the figures so the graphs could be enlarged.

---

## [Author Response · Author response to Decision Letter 1]

14 Jan 2020

Response to Reviewer’s comments

Their revision and written communication addresses some of the comments/questions raised by reviewers. They are careful to note their association between valve type is not causal. The clinical impact of this investigation is difficult to ascertain. The authors might include additional discussion on their interpretation of results – i.e. their interpretation of findings. They may have an idea on why, after risk adjustment, patients with calcific/degenerative pathology have worse outcomes (fatal and nonfatal complications) relative to patients with inflammatory pathology - especially given the strong association seen with adverse outcomes

Response: Because of potential residual confounding, we agree with the reviewer that the association which we found are most likely not fully causal. We do like to discuss potential reasons for the association. Based on our findings, we speculate that the valve etiology profiles might have some effect on mid-term outcomes even after the removal of the diseased valve. Possibly, those differences in late complications among three groups may stem from the extent of postoperative regression of diffuse myocardial fibrosis and myocardial cellular hypertrophy based on aortic valve etiology, as investigated in a recent study [1]. Recently, calcific and degenerative aortic stenosis (AS) has been understood to be an active disease process akin to atherosclerosis from the perspectives of compelling histopathologic and clinical data [2]. Our study results allow us to interpret that numerous overlaps in clinical factors related to calcific and degenerative valve disease and atherosclerosis might be a key finding to explain the potential mechanism by which patients in the calcific/degenerative group develop adverse events associated with plaque vulnerability and thrombosis, including new-onset ischemic heart disease and non-fatal stroke, even after surgical aortic valve replacement (AVR) with concomitant coronary artery bypass grafting (CABG) when indicated. Indeed, heart failure admission, unplanned coronary revascularization and cardiovascular death were more common in the calcific and degenerative compared to the other two groups. The systemic impact of these atherosclerotic disease may in part be considered a residual confounding when comparing the prognoses of surgical AVR patients with three different etiologies. However, it can also be considered a direct consequence of the AS that differently affects myocardium and its recovery by AS etiology. In contrast, congenital aortic valve disease often involves many vascular abnormalities [3], and post-inflammatory disease can affect the myocardium and heart valves and the brain, joints, and skin [4]. In our present study, 23% of patients in the congenital group had undergone replacement of ascending aorta concomitantly, and 82% and 36% of patients in the post-inflammatory group were added surgical procedures on mitral valve and tricuspid valve, respectively. These concomitant surgical procedures, when indicated, might show protective effect on clinical outcomes for potential late complications after surgical AVR. In other words, other valve abnormalities and aortic disease (aneurysm and dilatation) are more likely to coexist in both post-inflammatory and congenital AS patients, which could raise a risk of developing cardiovascular events. However, in most cases, these cardiac comorbidities are surgically treated concomitantly when indicated based on current guidelines, leading to risk reduction for potential cardiovascular events to occur after surgical AVR. On the other hand, patients with calcific/degenerative AS appear to have a strong association with systemic atherosclerosis and there might be an aspect that cardiovascular risks are inevitably left even though a concomitant CABG is properly performed.

Thus, we have strengthened the discussion on why patients with calcific/degenerative pathology have worse outcomes (fatal and nonfatal complications) relative to patients with inflammatory pathology (Page 26, Line 347): “Based on our findings, we speculate that the valve etiology profiles might have some effects on the mid-term outcomes even after the removal of the diseased valve. Possibly, those differences in late complications among the three groups may stem from the extent of postoperative regression of diffuse myocardial fibrosis and myocardial cellular hypertrophy based on the aortic valve etiology, as investigated in a recent study [27]. Recently, calcific and degenerative AS has been understood to be an active disease process akin to atherosclerosis from the perspectives of compelling histopathologic and clinical data [28]. Our study results allow us to interpret that numerous overlaps in clinical factors related to calcific and degenerative valve disease and atherosclerosis might be a key finding to explain the potential mechanism by which patients in the calcific/degenerative group develop adverse events associated with plaque vulnerability and thrombosis, including new-onset ischemic heart disease and non-fatal stroke, even after surgical AVR with concomitant CABG when indicated. Indeed, HF admission, unplanned coronary revascularization, and cardiovascular death were more common in the calcific and degenerative group than in the other two groups. The systemic impact of these atherosclerotic diseases may in part be considered a residual confounding when comparing the prognoses of surgical AVR patients with three different etiologies. However, it can also be considered a direct consequence of the AS that differently affects the myocardium and its recovery by the AS etiology. In contrast, the congenital aortic valve disease often involves many vascular abnormalities [29], and the post-inflammatory disease can affect the myocardium and heart valves and the brain, joints, and skin [30]. In our present study, 23% of patients in the congenital group had undergone replacement of ascending aorta concomitantly, and 82% and 36% of the patients in the post-inflammatory group underwent additional surgical procedures on the mitral and tricuspid valves, respectively. These concomitant surgical procedures, if indicated, might show a protective effect on the clinical outcomes for potential late complications after surgical AVR. In other words, other valve abnormalities and aortic diseases (aneurysm and dilatation) are more likely to coexist in both post-inflammatory and congenital AS patients, which could raise a risk of developing cardiovascular events. However, in most cases, these cardiac comorbidities are surgically treated concomitantly when indicated based on the current guidelines [7], leading to risk reduction of potential cardiovascular events that could occur after surgical AVR. On the other hand, patients with calcific/degenerative AS appear to have a strong association with systemic atherosclerosis, and there might be a possibility that cardiovascular risks inevitably remain even though a concomitant CABG is properly performed. Taking this into consideration, AS should be deemed as a systemic disease with different potential mechanisms by its valve etiology, although the definite pathophysiology behind AS remains incompletely investigated [31].”

Consider including the results of univariate analysis for the candidate variables use for multivariate analysis used in Table 4 (i.e.,, age, sex,....euroscore, etc)

Response: We have added the results of univariate analysis for all variables that were used for each multivariate analysis changing the format of Table 4. 

Also, In the discussion, the authors might briefly discuss recommendations on how their study findings should impact clinical practice.

Response: Our data suggest that the etiology and pathobiology of the valvular disease should be examined through a pathological investigation of the excised valve, which can provide essential information on its natural history, surgical risk, postoperative outcome, association with systemic disease, and potential vascular complications, in addition to preoperative imaging assessments. Therefore, future possibility of establishing accurate diagnosis of the diseased valves with less invasive imaging modalities confirmed by pathological examinations should be further explored, which can lead to another novel researches such as valve progression by aortic valve etiology or associations with non-aortic valves. In addition, our results show that the pathologically assessed etiology of the diseased valve was a strong predictor of the mid-term outcomes of the patient even after adjusting for patient demographic characteristics and perioperative risks (represented by EuroSCORE II). This could certainly guide clinicians and patients in decision-making regarding the postoperative follow-ups and support the importance of the accuracy of pathological valve diagnosis. More importantly, our data on detailed clinical outcomes highlights distinct trends in incidences of harmful events following surgical AVR among three etiologies. In fact, heart failure admission, device implantation and gastrointestinal bleeding were more prominent in both the congenital and calcific/degenerative groups compared to the post-inflammatory one while non-fatal stroke was more common in both the calcific/degenerative and post-inflammatory groups. In managing these patients, such a novel information could help to detect and treat potential late complications unique to each valve etiology. 

Thus, we reinforced the discussion section to clearly state clinical implications (Page 27, Line 382): “As a clinical implication, our data suggest that the etiology and pathobiology of the valvular disease should be examined through a pathological investigation of the excised valve, which can provide essential information on its natural history, surgical risk, postoperative outcome, association with systemic disease, and potential vascular complications, regardless of preoperative imaging assessments. Therefore, future possibility of establishing accurate diagnosis of the diseased valves with less invasive imaging modalities confirmed by pathological examinations should be further explored, which can lead to other novel researches, such as valve progression by aortic valve etiology or associations with non-aortic valves. Moreover, our data on the detailed clinical outcomes highlight distinct trends in the incidences of harmful events following surgical AVR among the three etiologies. For instance, HF admission, device implantation, and gastrointestinal bleeding were more prominent in both the congenital and calcific/degenerative groups than in the post-inflammatory group, whereas non-fatal stroke was more common in both the calcific/degenerative and post-inflammatory groups. In managing these patients, such novel information could help to detect and treat potential late complications unique to each valve etiology.”

Figures 3 and 4 are difficult to read. Consider reformatting the figures so the graphs could be enlarged.

Response: We have reformatted the figures 3 and 4 so that they could be more readable by enlarging the original graphs clearly. 

 

Reference

1. Treibel TA, Kozor R, Schofield R, Benedetti G, Fontana M, Bhuva AN, et al. Reverse Myocardial Remodeling Following Valve Replacement in Patients With Aortic Stenosis. J Am Coll Cardiol. 2018;71:860-71. doi: 10.1016/j.jacc.2017.12.035 PMID: 29471937

2. Freeman RV, Otto CM. Spectrum of calcific aortic valve disease: pathogenesis, disease progression, and treatment strategies. Circulation. 2005;111:3316-26. doi: 10.1161/CIRCULATIONAHA.104.486738 PMID: 15967862.

3. Siu SC, Silversides CK. Bicuspid aortic valve disease. J Am Coll Cardiol. 2010;55:2789-800. doi: 10.1016/j.jacc.2009.12.068 PMID: 20579534.

4. Marijon E, Mirabel M, Celermajer DS, Jouven X. Rheumatic heart disease. Lancet. 2012;379:953-64. doi: 10.1016/S0140-6736(11)61171-9 PMID: 22405798.

---

## [Editor Report · Decision Letter 2]

27 Jan 2020

PONE-D-19-19333R2

Clinical impact of pathology-proven etiology of severely stenotic aortic valves on mid-term outcomes in patients undergoing surgical aortic valve replacement

PLOS ONE

Dear Dr. Miura,

Thank you for submitting your manuscript to PLOS ONE. After careful consideration, we feel that it has merit but does not fully meet PLOS ONE’s publication criteria as it currently stands. Therefore, we invite you to submit a revised version of the manuscript that addresses the points raised during the review process.

The manuscript is improved and the reviewer's comments were addressed. However, the current discussion is too lengthy, with run-on sentences, redundant ideas and unnecessary repetition of the results.

Suggest to summarize the key results along with the main discussion points in a concise manner.  Limit the reviews of the topic to only what is pertinent to the discussion points. Please also review the final product with a professional English editor.

We would appreciate receiving your revised manuscript by Mar 12 2020 11:59PM. To enhance the reproducibility of your results, we recommend that if applicable you deposit your laboratory protocols in protocols.io, where a protocol can be assigned its own identifier (DOI) such that it can be cited independently in the future. For instructions see: http://journals.plos.org/plosone/s/submission-guidelines#loc-laboratory-protocols

We look forward to receiving your revised manuscript.

Kind regards,

Wen-Chih Wu, MD, MPH

Academic Editor

PLOS ONE

---

## [Author Response · Author response to Decision Letter 2]

7 Feb 2020

Response to Reviewer’s comments

The manuscript is improved and the reviewer's comments were addressed. However, the current discussion is too lengthy, with run-on sentences, redundant ideas and unnecessary repetition of the results. Suggest to summarize the key results along with the main discussion points in a concise manner. Limit the reviews of the topic to only what is pertinent to the discussion points. Please also review the final product with a professional English editor.

Response: As suggested by academic editors, we carefully summarized the main discussion points for improved readability and clarity and fully focused on the topic of our interests throughout the entire manuscript by avoiding redundancy and unnecessary ideas. Thus, we revised some parts, mainly in the discussion section, with our intended meanings retained that are highlighted in red ink so that you could follow the changes we made. This file was uploaded as a separate file and labeled 'Revised Manuscript with Tracked Changes' together with ‘Manuscript’ in the process of submitting the revised manuscript after the final product was carefully reviewed by a professional English editor.

---

## [Editor Report · Decision Letter 3]

13 Feb 2020

Clinical impact of pathology-proven etiology of severely stenotic aortic valves on mid-term outcomes in patients undergoing surgical aortic valve replacement

PONE-D-19-19333R3

Dear Dr. Miura,

We are pleased to inform you that your manuscript has been judged scientifically suitable for publication and will be formally accepted for publication once it complies with all outstanding technical requirements.

With kind regards,

Wen-Chih Wu, MD, MPH

Academic Editor

PLOS ONE
---

## [Editor Report · Acceptance letter]

26 Feb 2020

PONE-D-19-19333R3 

Clinical impact of pathology-proven etiology of severely stenotic aortic valves on mid-term outcomes in patients undergoing surgical aortic valve replacement 

Dear Dr. Miura:

I am pleased to inform you that your manuscript has been deemed suitable for publication in PLOS ONE. Congratulations! Your manuscript is now with our production department. 

With kind regards,

on behalf of

Dr. Wen-Chih Wu 

Academic Editor

PLOS ONE